# ‘Two in One’ Cloning Vector Applied for Blunt-End and T-A Cloning with One-Step Digestion–Ligation and Screening of Positive Recombinants by Unaided Eyes

**DOI:** 10.3390/cimb47010017

**Published:** 2024-12-31

**Authors:** Xingli Zhang, Chong Teng, Kaidi Lyu, Shanhua Lyu, Yinglun Fan

**Affiliations:** College of Agriculture and Biology, Liaocheng University, Liaocheng 252000, China; 2220190340@stu.lcu.edu.cn (X.Z.); 2110190206@stu.lcu.edu.cn (C.T.); 2210190201@stu.lcu.edu.cn (K.L.)

**Keywords:** cloning vector, PCR products, one-step cloning, digestion–ligation reaction, *mScarlet-I* gene

## Abstract

To clone DNA sequences quickly and precisely into plasmids is essential for molecular biology studies. Some cloning vectors have been developed for the cloning of PCR products, including blunt-end and T-A cloning. However, different plasmids are required for the cloning of PCR products with blunt ends and 3′ A overhang ends. Here, a novel cloning vector, pYFRed, which is based on the pUC19 backbone, has emerged and can be applied in both blunt-end and T-A cloning. PCR products can be cloned into the pYFRed by a one-step digestion–ligation reaction in a tube. The endonuclease recognition sequences of *Sma*I, *Eco*53kI, *Eco*RV, *Pme*I, and *Swa*I for blunt-end cloning and *Xcm*I for T-A cloning were designed and added between the *lac* promoter and the starting codon ATG of the *mScarlet-I* gene of pYFRed. The ligation efficiency was significantly higher because the restriction enzyme sites utilized were removed from the vector after being successfully constructed. The *mScarlet-I* gene was introduced into the pYFRed for the screening of the positive recombinants by the unaided eye without the need for additional reagents/equipment. pYFRed is easy to construct in an ordinary laboratory, which facilitates researchers to develop their cloning vector without purchasing commercial cloning vectors.

## 1. Introduction

It is usually necessary to clone PCR products to segregate DNA fragments when PCR amplification produces a complex mixture, acquires high-quality sequencing results, or obtains DNA fragments that can be preserved permanently [1]. Nowadays, CRISPR/Cas9 technology has become an essential tool for gene function verification [2]. After editing the target gene with CRISPR/Cas9 technology, it is often necessary to clone the target sequence into a cloning vector for DNA sequencing. At present, the commonly used DNA polymerases for PCR amplification are divided into highly thermostable DNA polymerase from *Thermus aquaticus* (Taq DNA polymerase) and high-fidelity DNA polymerase, according to its fidelity characteristics. When the Taq DNA polymerase, which has no 3′-exonuclease activity, is used for PCR amplification, adenosine is added to the 3′-end of the double-stranded DNA molecule [3]. Therefore, PCR amplification products can be cloned using a linear ‘T-vector’ with a single 3′-T overhang at both ends, and this cloning method is called T-A cloning [4]. DNA polymerases with high fidelity and 3′-5′ exonuclease activity, such as KOD DNA polymerases with high fidelity, produce blunt ends on PCR products after PCR amplification [5]. There are many reports on the development of vectors for the T-A cloning and blunt-end cloning of PCR products. In these reports, blunt-end and T-A cloning vectors are constructed separately and the *ccdB*, *lacZ*, or *gfp* genes are used as selection markers [6,7,8,9,10]. When these markers are used, a unique *E. coli* strain is needed for the *ccdB* gene; X-gal and IPTG are added in LB medium for the *lacZ* gene; and optical equipment is required in order to visualize *gfp* gene expression [10,11]. There are a few vectors that can be used for both T-A and blunt-end cloning. pXST was a novel vector for blunt-end and T-A cloning; however, pXST needed to be linearized when cloning PCR products [8]. pCRZeroT were designed for both the blunt-end and T-A cloning of PCR products, but there is only one *Sma*I restriction endonuclease site in pCRZeroT [9].

There are many methods for cloning foreign DNA fragments into a cloning vector. Despite the availability of numerous restriction-free cloning methods, restriction enzyme-based cloning techniques remain prevalent [12]. Most of the cloning vectors for PCR products were linearized with restriction endonuclease and dephosphorylated with dephosphorylase. The Golden Gate cloning method overcomes some shortcomings of the traditional restriction digestion and ligation method with a type IIs restriction endonuclease and T4 ligase in the same digestion–ligation reaction [13]. When utilizing the Golden Gate cloning method, the recognition sites of the restriction enzyme are removed from the vector after the foreign DNA fragment is successfully inserted. This cloning method effectively prevents the re-digestion of the ligated product, and improves the cloning efficiency. We are convinced that the ligation efficiency will be significantly higher if the type II restriction enzyme sites utilized are removed from the vector successfully constructed.

In this study, a ‘two in one’ cloning vector was developed with pUC19 as the basic vector and the *mScarlet-I* gene, coding a red fluorescent protein with higher brightness and making *E. coli* clones with red color visible by unaided eyes as the screening reporter [14]. The new cloning vector can be used for cloning PCR products with a blunt end and a 3′-A overhang. The new cloning vector, PCR products, restriction endonuclease, and T4 ligase can be added into one tube of reaction solution to complete the digestion of the vector and the ligation between the cloning vector and the PCR products. Experimental results demonstrate that the pYFRed vector exhibits high efficiency for the cloning of PCR products, as evidenced by the successful transformation and verification of recombinant colonies. According to the method of this study, the new cloning vector is easy to construct in an ordinary laboratory.

## 2. Materials and Methods

### 2.1. Strains, Reagents, and Cell Cultivation

Chemically competent cells, *Escherichia coli* DH5α, were purchased from TSINGKE Biotech (Qingdao, China) and used for heat shock transformation. Yeast extract (LP0021) and Tryptone (LP0042) were purchased from Thermo Scientific Oxoid and used to compound Lysogeny Broth (LB) medium for culturing *E. coli*. High-fidelity DNA polymerase, KOD-plus, was purchased from TOYOBO (Osaka, Japan). Taq PCR Master Mix (2×, with Blue Dye) carrying Taq DNA polymerase was purchased from Sangon Biotech (Shanghai) Co., Ltd. (Shanghai, China). Restriction endonucleases and T4 DNA ligase (400U/uL) were purchased from New England Biolabs (Ipswich, MA, USA). Primers were synthesized by TSINGKE Biotech (Qingdao, China). The vector pMRE-Tn-155 carrying *mScarlet-I* was ordered from Addgene (Addgene Plasmid #118537).

### 2.2. ‘All in One’ Cloning Vector with mScarlet-I Gene as Selecting Marker

The primer set, M199 and M198, were designed to amplify the *mScarlet-I* gene. Recognition sites of *Eco*RV (GAT↓ATC), *Pme*I (GTTT↓AAAC), and *Swa*I (ATTT↓AAAT) produced with a blunt end were added at the 5′-end of M199 (5′-GCGGAATTCGATATCCAAGTTTAAACTGGATTTAAATATGGTGAGCAAGGGCGAG-3′, the underlined sequence is the *Eco*RI recognition site for pYFRed construction). The sequence of the reverse primer M198 is 5′-GTGCACCATATGAAGCTTACTTGTACAGTTCGTCCATGCCTCCGGTGGAGTGGCGGC-3′ (the underlined sequence is the *Nde*I recognition site). PCR was amplified with KOD-plus polymerase and pMRE-Tn-155 plasmid as a template [15]. A PCR product digested with *Eco*RI and *Nde*I was cloned into pUC19 digested with *Eco*RI and *Nde*I with T4 ligase. The ligation mixture was transformed into chemically competent *E. coli* cells DH5α by 42 °C heat shock. The transformation mixtures were plated on the solid LB medium, which was not added with IPTG/X-gal (Isopropyl-beta-D- thiogalactopyranoside/5-Bromo-4-chloro-3-indolyl-beta-D-galactopyranoside). The clones with red fluorescence were picked to extract the plasmid DNA. The plasmid DNA was digested with *Eco*RI and *Nde*I to detect whether the newly constructed vector was correct. Sanger sequencing was also performed for verification.

### 2.3. One-Step Digestion–Ligation Reaction of Cloning Vector and PCR Products

If the DNA molecules of the PCR products are known to be identical and unique, the PCR products can be directly sequenced. However, when the DNA molecules at the gene-editing target site are diverse, it becomes necessary to clone the PCR products into a cloning vector to identify the specific types of gene editing. In our previous work, we established an efficient CRISPR/Cas9 gene editing system [16]. After gene editing, the edited target sequences needed to be cloned into a cloning vector for Sanger sequencing. Here, the edited target sequences of the *Rj7* gene of soybean were amplified with KOD-plus DNA polymerase with primer set *Rj71*/*Rj72* [16]. The 10 ng PCR amplification products without purification were mixed with 20 ng pYFRed, 10 U *Eco*RV, and 100 U T4 DNA ligase with 1 × NEB rCutsmart buffer with 1mM ATP in one tube of total 10 µL. The one-step restriction digestion–ligation reaction was performed in a thermo-cycler as follows: 10 cycles of 2 min at 37 °C, 5 min at 16 °C, 2 min at 12 °C, followed by a final incubated step for 5 min at 37 °C. The total of 10 µL ligation reaction solution was transformed into 100 µL *E. coli* DH5α cells with heat shock at 42 °C for 45 s, and the 100 µL transformation mixtures were added to 400 µL LB medium without antibiotic and then shaken in the incubator at 37 °C at 180 rpm for 1 h. Then, the 100 µL bacterial mixtures were cultured on the solid LB medium with 50 mg/L ampicillin for 14 h. The clones without red fluorescence were picked to extract plasmid DNA for digesting with *Hin*dIII and Sanger sequencing. Three biological replicates were independently performed for the digestion–ligation and transformation processes.

The edited target sequences of the *LjNLP1* gene of *Lotus japonicus* were amplified with Taq PCR Master Mix using primer set LjNLP1F/R [16]. The PCR products, pYFRed, *Xcm*I, and T4 DNA ligase were added into one tube and the following program was performed: 10 cycles of 2 min at 37 °C, 5 min at 16 °C, 2 min at 12 °C, and a final incubated step for 5 min at 37 °C. The transformation method was the same as above. The fifteen white clones and one red clone were directly used for PCR amplification with primer set LjNLP1F/R. The digestion–ligation and transformation processes were independently conducted three times.

## 3. Results

### 3.1. ‘All in One’ Cloning Vector pYFRed

The *mScarlet-I* gene (amplified with primers M199 and M198) was cloned between the *Eco*RI and *Nde*I sites of pUC19, replacing the open reading frame of the *lacZα* gene. This construction allowed the expression of the *mScarlet-I* gene under the control of the *lac* promoter. Surprisingly, there were many clones showing red fluorescence on LB medium without IPTG (Figure 1a), and displaying a red color which could be identified by the naked eye (Figure 1b). Five red clones and one white clone were randomly selected, and the plasmid DNA was extracted for digestion with *Hin*dIII and *Xcm*I. Agarose gel electrophoresis showed that the red-colored clones were positive transformants (Figure 1c). The new cloning vector was named as pYFRed (Figure 2). The cloning vector pYFRed in *E. coli* showed red-colored in LB medium without IPTG, which indicates that the *lac* promoter retained basal activity even in the presence of IPTG.

There are two blunt-end restriction endonuclease recognition sites, *Sma*I and *Eco*53kI, in the multi-cloning site (MCS) of pUC19. Another three blunt-end endonuclease recognition sites, *Eco*RV, *Pme*I, and *Swa*I (*Pme*I and *Swa*I are 8 bp restriction endonuclease), were introduced between the *lac* promoter and the start codon of the *mScarlet-I* gene. Thus, there were a total of five blunt-end restriction endonuclease recognition sites for cloning blunt-end PCR products (Figure 3). Another *Hin*dIII recognition site was also added into the pYFRed for the validation of PCR products by *Hin*dIII enzyme digestion (Figure 2).

For PCR products with 3′-A overhang cloning, there are two *Xcm*I endonuclease recognition sites, located in the start codon and termination codon, respectively, of *mScarlet-I* gene in pYFRed (Figure 3).

### 3.2. One-Step Digestion–Ligation of Blunt-End PCR Products with pYFRed

The edited target sequences of the soybean *Rj7* gene were cloned into pYFRed with one-step restriction-ligation. Eight white clones and one red clone were picked up to extract the plasmid DNA by alkali lysis. Plasmid DNAs were digested with restriction endonuclease *Hin*dIII for verification. The wild type of the *Rj7* sequence size was 591 bp. The sequence size between two *Hin*dIII sites in pYFRed was 771 bp (Figure 4). The fragment digested with *Hin*dIII from recombinant transformants was about 1362 bp because there was some insertion/deletion (In/Del) in the edited target sequence (Figure 4). The results showed that exogenous DNA fragment was correctly inserted into the recombinants of the white clones (Figure 4). The three ligations and transformations yielded 726,685, and 761 clones, with 622,587, and 645 being white-colored recombinants, respectively. On average, one transformation could yield 724 clones, of which 618 were positive clones. Without color selection, the cloning efficiency was 85%, while the efficiency for cloning blunt-end PCR products with pYFRed, using color selection, was 100%.

### 3.3. Cloning PCR Products with 3′-A Overhang into pYFRed

*Xcm*I endonuclease is required when preparing to clone a PCR product with a 3′-A overhang to the vector pYFRed. There are two *Xcm*I recognition sites in pYFRed. When the PCR product was inserted into pYFRed, the ORF of *mScarlet-I* was destroyed, and the color of the *E. coli* clones carrying recombinant transformants was white. The PCR product size of the edited target sequence in the *LjNLP1* gene of *L. japonicus* is 682 bp (Liu et al., 2022). In order to calculate the cloning efficiency of the pYFRed vector, fifteen white- and one red-colored clones were randomly selected for PCR analysis with the primer set LjNLP1F/R. The result showed 100% cloning efficiency as all the white clones had PCR products from the edited target sequences (Figure 5). The three transformations resulted in a total of 1320, 1231, and 1336 clones, of which 1165, 1098, and 1203 were white-colored recombinants, respectively. On average, a single transformation was capable of generating 1296 clones, with 1155 being positive clones. In the absence of color selection, the cloning efficiency reached 89%. The cloning efficiency of 3′ A overhang-end PCR products with pYFRed was 100% through color selection. The results suggest that the pYFRed vector was highly efficient in cloning PCR products with a 3′-A overhang.

### 3.4. Schematic Representation of Cloning PCR Products into pYFRed Using the One-Step Digestion–Ligation

When cloning PCR products into the pYFRed vector, it is not needed to linearize the vector with restriction endonuclease and dephosphorylate it with dephosphorylase. PCR products without purification can be used directly. The outline of PCR product-cloning into the pYFRed vector is shown in Figure 6. In the second step, the volume of PCR products is less than 2 µL and the pYFRed is about 20 ng. The molecular–molar ratio of PCR products to pYFRed should preferably be 1:1.

## 4. Discussion

Molecular cloning is the fundamental technology of modern molecular biology. Many molecular cloning methods have been developed such as restriction digestion- and ligation-based cloning, ligation-independent cloning (LIC), Infusion cloning, Gateway cloning, Gibson assembly, and Golden Gate cloning [17]. All these cloning methods can be divided into two types, namely, restriction digestion- and ligation-based cloning and homologous recombination cloning. Homologous recombination cloning methods, such as LIC, Gateway cloning, Infusion cloning, and Gibson assembly, rely on short homologous ends. When using a homologous recombination strategy for molecular cloning, it is necessary to purchase expensive commercial kits or kit components. There are also cloning methods that rely on homologous recombination that require more experimental steps, such as LIC. Using the Gateway cloning method, an attB recombinant sequence encoding eight amino acids will remain in the final recombinant vector. Based on the digestion–ligation method with type II restriction enzymes and ligase, DNA constructs are typically joined at restriction endonuclease sites and construction options are limited by the unique endonuclease sites available in the vector and gene. Because the recognition site of type II restriction endonuclease is their digestion site and the DNA sticky ends produced with type II restriction endonuclease are a palindromic sequence, the efficiency of digestion–ligation method is low. There was a novel cloning method, the Efficient Cloning Of Linear Inserts (ECOLI) technology, utilizing a site-directed mutagenesis, but it is restriction enzyme-free, not based on recombination [18].

The Golden Gate cloning method process approaches 100% efficiency because the required ligation products do not retain enzyme recognition sites, while the temporarily constructed destination vector and insert fragments retain functional Type IIS sites and will be re-digested [19]. When using the cloning vector pYFRed, the insertion of PCR products disrupts the restriction endonuclease recognition site, so the cloning vector and PCR products are highly efficiently assembled by the one-step digestion–ligation method. In this study, a single transformation using the cloning vector pYFRed yielded approximately 618 and 1155 positive recombinants when cloning PCR products with blunt-ends and 3′ A overhang ends, respectively. The efficiency of T-A cloning is 1.87 times higher than that of blunt-end cloning. However, due to factors such as the purity of vector plasmid DNA or the ratio of the vector to the exogenous fragment, the efficiency of this ligation may decrease. There are five blunt-end restriction sites, *Sma*I, *Eco*53kI, *Eco*RV, *Swa*I, and *Pme*I, for blunt-end ligation, and the provided PCR product does not contain the same restriction site. If the PCR product has one blunt-end restriction digestion site listed before, there are still four cloning sites to be used for cloning. If the sequence of the PCR product is unknown, we recommend using eight-base blunt-end restriction endonuclease *Swa*I, or *Pme*I. In pYFRed, the multi-cloning site of pUC19 was retained, and a digestion site of *Hin*dIII was added to detect the inserted fragment. Therefore, the pYFRed vector is very convenient for the detection of inserted DNA fragments.

Using the pYFRed vector not only enhances cloning efficiency, but also reduces the cost of use. According to the method of this study, this cloning vector is also easy to construct in an ordinary laboratory, which facilitates researchers to build their own cloning vector without purchasing commercial cloning vectors. IPTG and X-gal need to be added to the LB medium to promote the activity of the *lac* promoter in pUC18 series vectors for blue–white screening [20]. Red fluorescent protein translated by *mScarlet-I* gene can be seen in the clones of the pYFRed vector in LB medium without IPTG and X-gal, which indicates that the working of the *lac* operator is not strictly regulated by IPTG and that the *lac* promoter can still start gene expression without IPTG induction. The minimal promoter activity of the *lac* promoter is enough to make the *mScarlet-I* gene express red fluorescent protein visible by the naked eye. Consequently, the pYFRed vector reduces the cloning cost. The gene *mScarlet-I* in pYFRed is drived by the *lac* promoter, and glucose is a known repressor of the *lac* operon [21]. Adding glucose with a concentration exceeding 0.25 mmol/L as a supplement to LB medium inhibited the activity of the *lac* promoter in pYFRed, and the *E. coli* clone did not emit red fluorescence (Appendix A), which prevented the positive screening of transformants. Therefore, when using the pYFRed vector for positive clone selection, the glucose content in the culture medium cannot exceed 0.25 mmol/L, otherwise it will not be possible to screen by color.

## Figures and Tables

**Figure 1 cimb-47-00017-f001:**
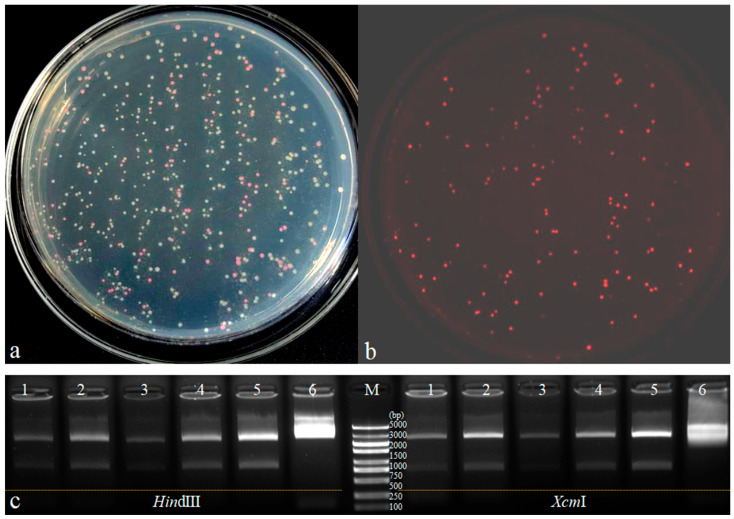
Transformed *E. coli* in LB medium without IPTG. (**a**): transformed clones under natural light; (**b**): detected with Tanon-5200Multi machine (Tanon Co., Ltd., Shanghai, China) with green excitation at 540 nm and emission at 600 nm; (**c**): agarose gel electrophoresis image with five red-colored clones (Lane 1–5) and white-colored clone (Lane 6) digestion with *Hin*dIII and *Xcm*I. Lane M: DL5000 DNA marker.

**Figure 2 cimb-47-00017-f002:**
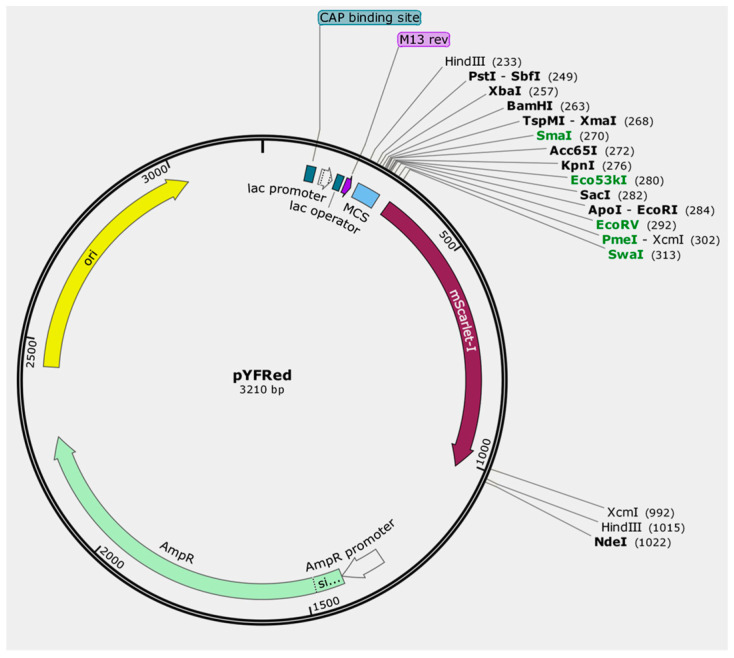
A diagrammatic presentation of the pYFRed cloning vector. The basic skeleton, including the AmpR-ori-*lac* promoter cassette, of the vector comes from the pUC19 vector. The blunt-end restriction endonuclease recognition sites are shown in green and were used for blunt-end cloning. Two *Xcm*I digested sites were used for T-A cloning. The *mScarlet-I* gene is marked in red.

**Figure 3 cimb-47-00017-f003:**
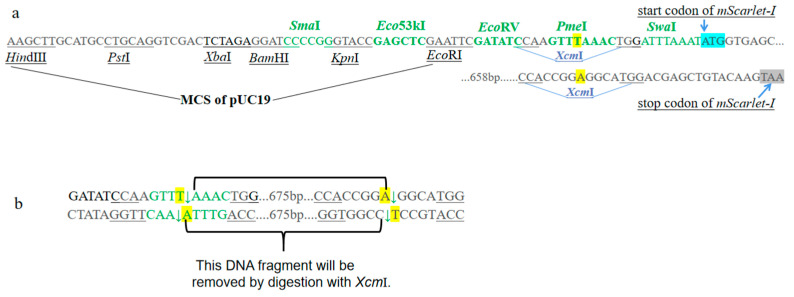
The cloning site regions of pYFRed. (**a**): the restriction sites for blunt-end cloning are indicated with green letters, and the restriction sites for T-A cloning are shown with blue letters. (**b**): pYFRed will be digested by *Xcm*I, and the produced linear DNA molecule would have a T-overhang in the 3′-end (the yellow-highlighted letter T in the first *Xcm*I digested site will be retained and the yellow-highlighted letter A in the second *Xcm*I digested site will be removed in this strand). The blue arrows show the start (ATG with blue highlight), and stop (TAA with grey highlight) codons of *mScarlet-I*.

**Figure 4 cimb-47-00017-f004:**
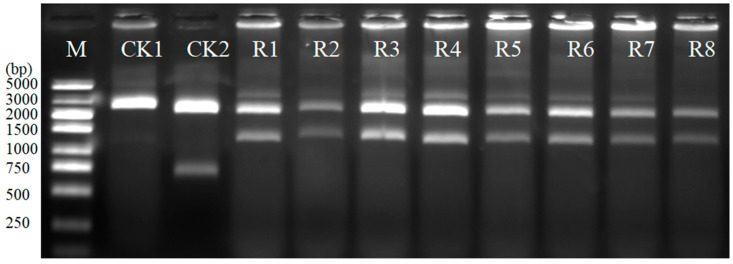
Enzyme digestion with *Hin*dIII of recombinants. M: DL5000 DNA marker; CK1: pUC19; CK2: pYFRed; R1-R8: white clones of recombinant transformants.

**Figure 5 cimb-47-00017-f005:**
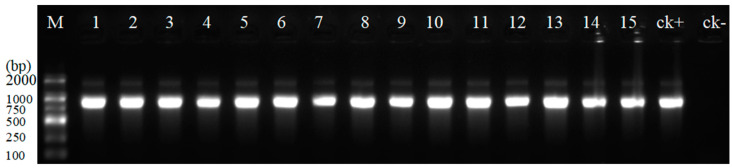
PCR assay of transformed clones. M: a DL2000 DNA marker; 1–15: white clones; ck+: positive control (soybean DNA); ck−: negative control (red clone).

**Figure 6 cimb-47-00017-f006:**
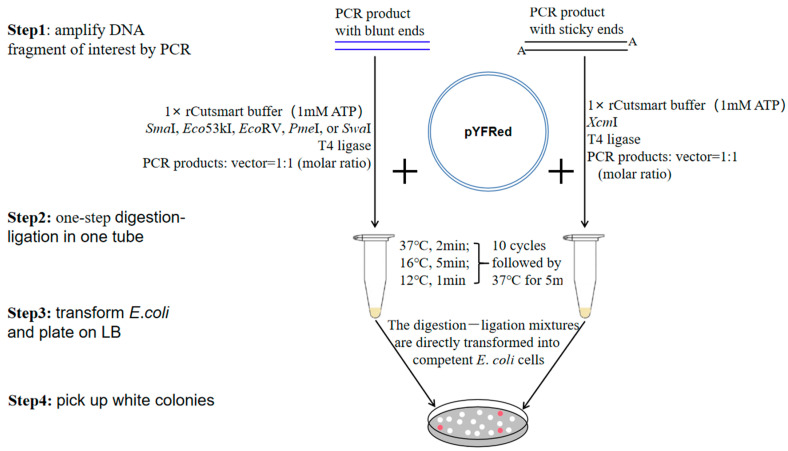
Cloning strategy of PCR product-cloning into pYFRed vector. DNA fragment of interest and pYFRed vector are mixed in one tube together with blunt-end restriction endonuclease or *Xcm*I and T4 ligase. The one-step digestion–ligation mixture is transformed into *E. coli*. All white clones are positive clones.

## Data Availability

All data supporting the conclusions of this article are included in this article. The GenBank accession number of pYFRed is PQ778476.

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
