# Peer review of "‘Two in One’ Cloning Vector Applied for Blunt-End and T-A Cloning with One-Step Digestion–Ligation and Screening of Positive Recombinants by Unaided Eyes"

_cimb, 2024, doi:10.3390/cimb47010017_

Round 1
Reviewer 1 Report
Comments and Suggestions for Authors
In this manuscript, the authors constructed a plasmid system included both blunt-end and T-A cloning for quick DNA sequence assembly. This system is convenience. However, more data are needed to be supplied.
1. Avoid using “many colonies” (L125). Give the extract numbers and calculate the rate, or just focus on the key plasmid pYFRed.
2. In section 3.2/3.3/3.4, at least enzyme digestion of 10 colonies (white or red ) needed to be provided. Calculate the positive rate and compared among three different methods.
3. In section 3.2/3.3/3.4, count the amounts of colonies in red or white and calculate the positive rate (positive colonies/amounts of competent cells), at least triplicate. Compared the rate among three different methods and give more discussion.
4. More discussion needed to be provided. Compared the details of this system with others, such as how many hours you saved for finishing constructing one plasmid and the material costs. The potential application of this system. Give more discussion to supply the word “efficiently”
5. Tons of mistakes. Read this manuscript carefully and revised it. L193: should be 3.4; L125 colonies; and many others.
Author Response
- Avoid using “many colonies” (L125). Give the extract numbers and calculate the rate, or just focus on the key plasmid pYFRed.
Response: Thank you for point this out. The ‘fifteen white clonies’ was reversed in L130 (before L125).
- In section 3.2/3.3/3.4, at least enzyme digestion of 10 colonies (white or red ) needed to be provided. Calculate the positive rate and compared among three different methods.
Response: Thanks your suggestion. In section 3.2., A total of eight Plasmid DNAs were digested with restriction endonuclease Hindâ…¢ for verification. And the sentence ‘and the cloning efficiency of blunt-end PCR product with pYFRed is 100%’ was added in the end of section 3.2. In section 3.3., A total of fifteen clones used for PCR analysis with primer set LjNLP1F/R, not digestion.
- In section 3.2/3.3/3.4, count the amounts of colonies in red or white and calculate the positive rate (positive colonies/amounts of competent cells), at least triplicate. Compared the rate among three different methods and give more discussion.
Response: Thanks your suggestion. We counted the number of recombinants with three independent transformations. In our lab, the vector pYFRed were used to clone PCR DNA fragment at least ten times and a single transformation yielded approximately about 600, and 1100 positive recombinants with blunt-ends, and 3’ A overhang ends, respectively. The amounts of colonies were counted (In Lane188-193, and Lane206-211). The cloning efficiency reach 100% through clone color seclection. More discussion was in Lane251-255.
- More discussion needed to be provided. Compared the details of this system with others, such as how many hours you saved for finishing constructing one plasmid and the material costs. The potential application of this system. Give more discussion to supply the word “efficiently”
Response: Thank your suggestion. We revised the discussion and added discussion for cloning efficiency in Lane251-263.
- Tons of mistakes. Read this manuscript carefully and revised it. L193: should be 3.4; L125 colonies; and many others.
Section 3.3 in L200 was revised to 3.4.
Response: Thank you for point this out. Lots of mistakes were revised.
Reviewer 2 Report
Comments and Suggestions for Authors
To authors
This manuscript introduces pYFRed, a novel “two-in-one” cloning vector designed to streamline the cloning of PCR products with both blunt ends and TA overhangs. Based on the pUC19 backbone, the vector incorporates the mScarlet-I gene as a visual marker, enabling the screening of recombinant colonies by eye without additional reagents.
I believe this tool holds significant potential for researchers looking for cost-effective and versatile cloning solutions, especially in resource-limited settings. While I have a few technical questions aimed at clarifying aspects of the digestion-ligation protocol, my primary concern is the need for a thorough revision of the English throughout the manuscript to ensure clarity and precision.
Below are my detailed comments:
Technical points:
1. Could the authors clarify the following details regarding the reaction mixture (in section 2.3)?
1.1. What is the total reaction volume used?
1.2. What is the typical amount of DNA present in the “2 µL PCR amplification products” mentioned in line 107?
1.3. What is the typical amount of pYFRed used? (later you state in results that you use 20 ng but this should be clear in methods)
1.4. What is the typical amount of digestion-ligation reaction used for transformation per volume of cells? Additionally, does this volume differ depending on whether blunt-end or T-A cloning is performed?
2. Since mScarlet-I expression is under the control of the lac promoter, and glucose is a known repressor of the lac operon, it would be helpful to mention in the Discussion that the addition of glucose as a supplement in bacterial experiments might suppress red fluorescence, potentially preventing the positive screening of transformants.
3. The authors mention using rCutsmart buffer with 1 mM ATP for the digestion-ligation reaction (in Methods and Figure 6). It would be helpful to explicitly state in the text that ATP is essential for the ligation step, as it provides the necessary energy for T4 DNA ligase activity.
4. State in text what is the pYFRed copy number.
5. The authors should provide the nucleotide sequence of pYFRed, either as supplementary material or by submitting it to a public gene database with an accession number.
Below, I detail some specific language-related issues and provide examples for improvement.
English points:
The manuscript would benefit from careful editing to address grammatical issues and improve overall style and clarity. Below are a few examples with suggested modifications for reference.
Lines 55-57: “When using the Golden Gate cloning method, the endonuclease recognition site of cloning vector is eliminated when foreign DNA fragment IS cloned into the vector, so the cloning efficiency is very high.”
This statement is not entirely clear of how the elimination of the recognition site enhances the efficiency of the cloning process in its current form. I suggest rewriting it as:
“When using the Golden Gate cloning method, the recognition site for the restriction enzyme is removed from the vector after the foreign DNA fragment is successfully inserted. This prevents re-digestion of the ligated product, thereby improving the cloning efficiency.”
Lines 67-68: “Therefore, the cloning vector has very high efficiency for cloning with PCR products.”
This sentence, in its current form, is not clear (given the previous text). I suggest the authors rewrite it as: “Experimental results demonstrate that the pYFRed vector exhibits high efficiency for the cloning of PCR products, as evidenced by the successful transformation and verification of recombinant colonies.”
Lines 100-103: This part needs revision for clarity. Perhaps something like:
“If the DNA molecules of the PCR products are known to be identical and unique, the PCR products can be directly sequenced. However, when the DNA molecules at the gene-editing target site are diverse, it becomes necessary to clone the PCR products into a cloning vector to identify the specific types of gene editing.”
Lines 123-135: This section requires particular attention, as it is currently confusing and contains redundant information (seems that information is duplicated).
Maybe the authors could consider starting with: “The mScarlet-I gene (amplified with primers M199 and M198) was cloned between EcoRI and NdeI sites of pUC19, replacing the open reading frame of the lacZα gene. This construct allowed expression of the mScarlet-I gene under the control of the lac promoter.” Then you could go to the “surprising” part, where you explain that the lac promoter retains basal activity even in the presence of IPTG. This approach not only clearly identifies the cloned fragment at the beginning but also allows for the removal of lines 131-135, eliminating redundancy.
Check and correct (note that this list is not exhaustive):
a. Italics in species names: e.g. Thermus aquaticus (line 32); Escherichia coli (line 72)
b. Italics in endonuclease names (e.g. SmaI in line 48 and several other places)
c. Italics in “lac” promoter (e.g. line 15)
d. “E.coli” to “E. coli” (several places throughout the manuscript)
e. Gene names (change “LacZ” to “lacZ”, “GFP” to “gfp”, etc etc)
f. “blent-end” to “blunt-end” (e.g. lines 147-148)
g. “there were two XcmI” to “there are two XcmI” (line 156)
h. “two HindIII” to “two HindIII sites” (line 171)
i. “will be remained in the finished recombinant vector” to “will remain in the final recombinant vector” (lines 218-219)
Comments on the Quality of English Languagestated above in the "comments for authors" section.
Author Response
Technical points:
- Could the authors clarify the following details regarding the reaction mixture (in section 2.3)?
1.1. What is the total reaction volume used?
1.2. What is the typical amount of DNA present in the “2 µL PCR amplification products” mentioned in line 107?
1.3. What is the typical amount of pYFRed used? (later you state in results that you use 20 ng but this should be clear in methods)
1.4. What is the typical amount of digestion-ligation reaction used for transformation per volume of cells? Additionally, does this volume differ depending on whether blunt-end or T-A cloning is performed?
Response: Thank your suggestion. The technical points were revised in the revised version.
- Since mScarlet-I expression is under the control of the lacpromoter, and glucose is a known repressor of the lac operon, it would be helpful to mention in the Discussion that the addition of glucose as a supplement in bacterial experiments might suppress red fluorescence, potentially preventing the positive screening of transformants.
Response: Thank you for point this out. This a good question. So we added one experiment in ‘2.4. The effect of glucose on the expression of mScarlet-â… ’ and ‘3.4. Glucose inhibits the expression of mScarlet-â… ’. In the discussion, the following sentences were added: ‘Adding glucose with a concentration exceeding 0.25 mmol/L as a supplement to LB medium inhibited the activity of the lac promoter in pYFRed, and the E. coli clone did not emit red fluorescence (Supplementary Figure1), which prevented the positive screening of transformants. Therefore, when using pYFRed vector for positive clone selection, the glucose content in the culture medium cannot exceed 0.25 mmol/L, otherwise it will not be able to screen by color.’ And we added one Supplementary Figure1 for the effect of glucose on the expression of mScarlet-â… of pYFRed.
- The authors mention using rCutsmart buffer with 1 mM ATP for the digestion-ligation reaction (in Methods and Figure 6). It would be helpful to explicitly state in the text that ATP is essential for the ligation step, as it provides the necessary energy for T4 DNA ligase activity.
Response: Yes. Thank you.
- State in text what is the pYFRed copy number.
Response: Thank you for point this out. The pYFRed is derived from pUC19, and we speculate that the copy numbers of these two plasmids are the same. Because during plasmid extraction, the amount of plasmid DNA extracted from the same volume of E.coli harboring pYFRed and pUC19 is the same (Figure 4).
- The authors should provide the nucleotide sequence of pYFRed, either as supplementary material or by submitting it to a public gene database with an accession number.
Response: Yes, the sequence was submitted to NCBI and the GenBank accession number is PQ778476 (BankIt2905593 pYFRed PQ778476). The information was added in Data Availability Statement.
Below, I detail some specific language-related issues and provide examples for improvement.
English points:
The manuscript would benefit from careful editing to address grammatical issues and improve overall style and clarity. Below are a few examples with suggested modifications for reference.
Lines 55-57: “When using the Golden Gate cloning method, the endonuclease recognition site of cloning vector is eliminated when foreign DNA fragment IS cloned into the vector, so the cloning efficiency is very high.”
This statement is not entirely clear of how the elimination of the recognition site enhances the efficiency of the cloning process in its current form. I suggest rewriting it as:
“When using the Golden Gate cloning method, the recognition site for the restriction enzyme is removed from the vector after the foreign DNA fragment is successfully inserted. This prevents re-digestion of the ligated product, thereby improving the cloning efficiency.”
Response: Thank you so much. We accepted your revision.
Lines 67-68: “Therefore, the cloning vector has very high efficiency for cloning with PCR products.”
This sentence, in its current form, is not clear (given the previous text). I suggest the authors rewrite it as: “Experimental results demonstrate that the pYFRed vector exhibits high efficiency for the cloning of PCR products, as evidenced by the successful transformation and verification of recombinant colonies.”
Response: Thank you so much. We accepted your revision.
Lines 100-103: This part needs revision for clarity. Perhaps something like:
“If the DNA molecules of the PCR products are known to be identical and unique, the PCR products can be directly sequenced. However, when the DNA molecules at the gene-editing target site are diverse, it becomes necessary to clone the PCR products into a cloning vector to identify the specific types of gene editing.”
Response: Thank you so much. We accepted your revision.
Lines 123-135: This section requires particular attention, as it is currently confusing and contains redundant information (seems that information is duplicated).
Maybe the authors could consider starting with: “The mScarlet-I gene (amplified with primers M199 and M198) was cloned between EcoRI and NdeI sites of pUC19, replacing the open reading frame of the lacZα gene. This construct allowed expression of the mScarlet-I gene under the control of the lac promoter.” Then you could go to the “surprising” part, where you explain that the lac promoter retains basal activity even in the presence of IPTG. This approach not only clearly identifies the cloned fragment at the beginning but also allows for the removal of lines 131-135, eliminating redundancy.
Response: Thank you so much. We accepted your revision.
Check and correct (note that this list is not exhaustive):
- Italics in species names: e.g. Thermus aquaticus(line 32); Escherichia coli (line 72)
- Italics in endonuclease names (e.g. SmaI in line 48 and several other places)
- Italics in “lac” promoter (e.g. line 15)
- “E.coli” to “E. coli” (several places throughout the manuscript)
- Gene names (change “LacZ” to “lacZ”, “GFP” to “gfp”, etc etc)
- “blent-end” to “blunt-end” (e.g. lines 147-148)
- “there were two XcmI” to “there are two XcmI” (line 156)
- “two HindIII” to “two HindIII sites” (line 171)
- “will be remained in the finished recombinant vector” to “will remain in the final recombinant vector” (lines 218-219)
Response: Thank you so much. All these were revised .